# Active Commuting to University Is Positively Associated with Physical Activity and Perceived Fitness

**DOI:** 10.3390/healthcare10060990

**Published:** 2022-05-26

**Authors:** Ximena Palma-Leal, Maribel Parra-Saldías, Salomé Aubert, Palma Chillón

**Affiliations:** 1PROFITH “PROmoting FITness and Health through Physical Activity” Research Group, Sport and Health University Research Institute (iMUDS), Department of Physical Education and Sports, Faculty of Sport Sciences, University of Granada, 18011 Granada, Spain; ximena.palmaleal@gmail.com; 2IRyS Group, School of Physical Education, Pontificia Universidad Católica de Valparaíso, Viña del Mar 2340000, Chile; 3Departamento de Educación Física, Deporte y Recreación, Universidad de Atacama, Copiapó 1532297, Chile; maribel.parra@uda.cl; 4Active Healthy Kids Global Alliance, Ottawa, ON K1H 8L1, Canada; salome_aubert@hotmail.fr

**Keywords:** fitness, physical activity, active commuting, university students, health behaviors

## Abstract

Background: Fitness is a powerful marker of health associated with physical activity (PA) in university students. However, insufficient PA is a serious health concern among university students. Active commuting provides an opportunity for increased PA levels. Therefore, the aims of this study were (a) to describe the mode of commuting, PA and fitness in university students; (b) to analyze the associations of mode of commuting with PA and fitness; and c) to analyze the relationship between mode of commuting, PA recommendations and fitness. Methods: This was a cross-sectional study. A total of 1257 university students (52.4% women) participated (22.4 ± 5.6 years old). Results: Public commuting was the main mode to and from university. Active and public commuters were more likely to meet the PA recommendations and reported higher muscular strength than those using private commuting. Active and public commuters who met PA recommendations present the highest fitness in most of its components. Conclusions: Achieving the PA recommendations was more relevant than adopting an active mode of commuting in order to have better fitness. Further research targeting a broader understanding of the mode of commuting, PA levels and fitness in university students is needed.

## 1. Introduction

There is irrefutable evidence that physical activity (PA) reduces the risk of premature mortality and is an effective primary and secondary preventive strategy for at least 25 chronic medical conditions [1]. The PA recommendations for adults developed by the World Health Organization (WHO) indicate that adults should engage in a combination of 75–150 min of moderate and vigorous PA (MVPA) throughout the week to obtain substantial health benefits [2]. However, the majority of adults worldwide do not meet the recommended amount of PA, and this trend appears to continue over time [3].

An insufficient level of PA is a serious health concern among university students. This specific population was described as emerging adults [4,5] who are simultaneously in positions of vulnerability and influence as they enter a new stage in life that includes more responsibility while being exposed to increased opportunities for personal growth [6]. Similar to adults, the unfavorable patterns of PA are evident in university students [7]. In Canada, more than one-half of the university students were estimated to not meet sufficient levels of PA [8]. In Spain, only 5.4% of university students were estimated to meet the recommendation of MVPA [9]. In the United States, fewer than 10% of university students were estimated to meet 30 min of MVPA 5 days a week [10], and in Chile, every year, there is an observed decrease in the levels of PA in university students [11]. Therefore, there is an urgent need to invest directly in strategies to promote PA among university students.

The main health message directed to all populations is that doing some PA is better than doing none [2,12]. Active commuting (AC), such as walking or cycling, provides an opportunity for increased PA levels in university students by increasing energy expenditure per minute [13]. In addition, a systematic review demonstrated that children and adolescents who walk or cycle to school can accumulate between 5 and 37 additional minutes of PA per day compared to those using passive modes of commuting [14]. However, students starting university tend to decrease their AC compared to when they were in secondary education [15,16].

AC can play an important role in addressing the issue of insufficient level of PA in university students. Indeed, in Latin American countries, such as Chile, associations with AC could be important and useful for public health, by the fact that recommending this active behavior in daily routines could be an important contributor to PA levels, which would be reflected in health benefits in the university population, known to be an inactive population with unhealthy habits. In addition, increased PA levels have been associated with greater fitness in university students [17]. Fitness is a powerful marker of health [18] that has been previously studied among university students. Studies showed that students from Qatar with greater fitness had lower morbidity and mortality [19], while students from Colombia had a healthy blood pressure levels, fat-free mass index and triglyceride levels and a lower prevalence of obesity indicators, even if it was self-reported fitness [20]. Therefore, associations between AC and PA, as well as fitness, may be of high interest in detecting the physical health benefits that active modes of commuting may contribute in this population and future programs to be implemented at universities, especially in Latin American countries, which are difficult to compare with developed countries, such as the Netherlands or Germany, where the use of active commuting is a common part of the university routines. Therefore, the aims of the current study were (a) to describe the mode of commuting, PA and fitness in university students by sex; (b) to analyze the associations of mode of commuting with PA and fitness by sex; and (c) to analyze the relationship between mode of commuting, MVPA recommendations and fitness by sex.

## 2. Materials and Methods

### 2.1. Study Design and Participants

This cross-sectional study was conducted between April and November of 2017 during normal class periods. A total of 1257 university students (52.4% women) with an average age of 22.4 ± 5.6 years participated in this study. The participants were recruited from three different public and private universities (public: Pontificia Universidad Católica de Valparaíso and Universidad Técnica Federico Santa María; private: Universidad de las Américas) located in urban cities in the central zone of Chile (two in Valparaíso and one in Santiago). The students belonged to diverse faculties (art, engineering, health, social sciences and education), and their length of study ranged from one to ten semesters. The sampling was via convenience. The students all agreed to participate voluntarily in the study, recruited from special courses of “healthy life” or “recreational program of physical activity”. Those students enrolled between 2010 and 2015, as well as those belonging to the career of physical education or athletes involved in the university leagues, were excluded.

### 2.2. Procedures and Ethical Requirements

A letter with the objectives of the study was sent to the corresponding authorities of the selected universities that were chosen by convenience because there were local teachers interested in this research. Once the universities’ authorizations were obtained, all university students attending the above courses were invited to voluntarily participate in the present study by means of a face-to-face presentation where they received detailed information about the objectives and methods of the study. Those who agreed to participate in the study completed a letter of informed consent in which the characteristics of the questionnaire, the purpose of the study and the confidentiality of the results were explained again. Students completed a 15 to 30 min self-reported paper-based questionnaire that was distributed and guided by the previously trained volunteer local teacher researchers. All procedures followed the Helsinki protocols [21] and were approved by the Ethics Committee of Pontificia Universidad Católica de Valparaíso (Code: CCF02052017).

### 2.3. Instruments

The self-reported questionnaire used for this study was created at the School of Physical Education by researchers with expertise on the topic of AC and was called “Questionnaire of mode of commuting and PA to the university”. It was created based on a literature review and consultations with experts. This questionnaire includes questions about sociodemographic variables and commuting behaviors. Every question was adapted to university students’ context and was found reliable for Chilean university students [22].

### 2.4. Mode of Commuting

The question about the mode of commuting was selected based on a systematic review of 158 peer-reviewed publications as the most appropriate measurement for assessing the mode of commuting by self-reported questionnaire [23]. This specific question was found to have an appropriate validity [24] and reliability [25] in young Spanish people and was found reliable for university students [22]. The mode of commuting to and from university was assessed using separate questions: How do you usually travel to university? How do you usually travel from university? The answer options were walking, cycling, car, motorcycle, public bus, metro/train and others. Participants were classified into three categories: “active” (walking and cycling), “private” (car and motorcycle) and “public” (public bus and metro/train) commuting [26]. Students with combined answers (e.g., active + private) were classified in the mode of commuting involving the highest PA levels. AC involves the highest PA levels, followed by public commuting which involves an intermediate level of PA in walking to and from stations and stops; private commuting is assumed to involve the lowest PA levels [27].

### 2.5. PA and Sedentary Behavior

PA was evaluated using the seven questions of the International PA Questionnaire (IPAQ) short version [28]. The IPAQ assesses three types of PA (light, moderate and vigorous) and sedentary behavior. The final output provides the sum of the duration (in minutes) and frequency (days) of light, moderate, vigorous and total PA and sedentary behavior. Participants’ PA level was calculated and categorized using metabolic equivalents (METs): light PA level corresponded to 3.3 METs × minutes × days per week; moderate PA level to 4 METs × moderate minutes × days per week; and vigorous PA level to 8 METs × minutes × days per week. Total PA corresponded to the sum of light, moderate and vigorous PA. Participants were also classified according to the MVPA recommendations for adults (≥150 min/week) as meeting MVPA recommendations or not meeting MVPA recommendations [12]. In addition, sedentary behavior was expressed in hours per day (h/day).

### 2.6. Fitness

The perceived fitness was self-reported using the International Fitness Scale (IFIS) based on the answers to six basic questions about fitness: general physical condition, cardio-respiratory fitness, muscular strength, speed and agility, flexibility and general health [29]. The possible Likert-scale answers, on a five-point scale, were “very poor” (1), “poor” (2), “average” (3), “good” (4) and “very good” (5) [29]. The mean of each fitness component was calculated and used for the analysis.

### 2.7. Statistical Analysis

Descriptive statistics were reported for participants’ mode of commuting, PA and fitness, separated by gender (men or women). Means and standard deviation (SD) were reported for continuous variables, and frequencies and percentages (%) were reported for categorical variables. The significant differences in these descriptive variables for men and women were analyzed using chi-square test for categorical variables and standard analysis of variance (ANOVA) for continuous variables, where the level of significance was set to *p* < 0.05. Associations between mode of commuting and PA, as well as fitness, were studied using multinomial logistic regression analysis. Mode of commuting was included in the model as the dependent variable, where private commuting was established as a reference, and PA and fitness were included as independent variables in separate models. We conducted these analyses separately for men and women. A complementary analysis was repeated conducting the same multinomial logistic regression but adjusting for fitness and PA in order to take into account the potential interaction between PA and fitness. In addition, another complementary analysis was carried out to identify the associations between PA recommendations and fitness by gender. PA recommendations were included in the model as the dependent variable, where not meeting MVPA recommendations was established as a reference, and fitness was included as an independent variable. All logistic regression analyses were separated by gender. An additional analysis was conducted to examine the effect of both mode of commuting and PA on fitness. For this analysis, six categorical groups based on the three modes of commuting (active, public and private) and the PA (meeting or not meeting MVPA recommendations) were calculated (e.g., active–meeting MVPA, public–meeting MVPA, private–meeting MVPA, active–not meeting MVPA, public–not meeting MVPA and private–meeting MVPA). All of the six fitness questions (general physical condition, cardio-respiratory fitness, muscular strength, speed and agility, flexibility and general health) were included as independent variables in a one-way analysis of variance ANOVA, and a post hoc subcommand Bonferroni test was used to compare the means of the categorical groups created between the three modes of commuting and PA (meeting or not meeting MVPA recommendations). This analysis was adjusted by gender. The statistical analyses were conducted using IBM SPSS Statistics (v. 25.0 for Windows, Chicago, IL, USA).

## 3. Results

Mode of commuting, PA, sedentary behavior and fitness of the participants, as well as the significant differences by gender, are presented in Table 1. The main mode of commuting to and from university was public modes in men (55.5%) and women (69.7%), followed by active modes (men: 34.4% (walking = 32.6%, cycling = 1.8%); women: 19.0% (walking = 17.5%, cycling = 1.5%)) and private modes (men: 10.0%; women: 11.4%) (*p* < 0.001). Regarding PA levels, most men showed a vigorous PA level (43.3%), while most women showed a light PA level (49.6%) (*p* < 0.001). In addition, 54.5% of men and 41.4% of women (*p* < 0.001) reported meeting the MVPA recommendations. The mean total PA time was 1.9 ± 1.8 h/day in men and 1.2 ± 1.4 h/day in women (*p* < 0.001). In contrast, the mean sedentary behavior time was 6.8 ± 6.9 h/day in men and 6.2 ± 5.9 in women (*p* < 0.001). There were significant differences among all fitness components (*p* < 0.001), where women presented lower mean scores, except for flexibility.

The associations of mode of commuting with PA levels, sedentary behavior, and fitness are presented separately for gender in Table 2. Active commuters had higher moderate (men: *p* = 0.021) and vigorous PA levels (women: *p* = 0.022) than private commuters. In addition, active commuters met the MVPA recommendations (men: *p* = 0.042; women: *p* = 0.028) and had higher total PA (women: *p* = 0.036) than private commuters. On the other hand, active and public commuters reported engaging in less sedentary behavior (*p* < 0.001, *p* = 0.001, respectively) than those who used a private mode of commuting. Regarding fitness, male active and public commuters showed higher muscular strength (*p* = 0.001, *p* = 0.003, respectively) compared to those who used a private mode of commuting. Finally, public commuters reported higher flexibility and general health levels (women: *p* = 0.030; men: *p* = 0.040, respectively) compared to students who used private modes of commuting.

The same analysis was repeated adjusting by PA recommendations and fitness (except in the analysis where that variable was the predictor variable), and similar results were obtained. Male public commuters had higher moderate and vigorous PA levels and met the MVPA recommendations more than private commuters. Concerning fitness, female active commuters showed higher muscular strength and flexibility compared to those who used a private mode of commuting (Appendix A). Regarding the association between PA and fitness, men and women who met the MVPA recommendations had higher fitness (all *p* < 0.001) than those who did not meet the MVPA recommendations (Appendix A).

The associations of the mode of commuting and meeting MVPA recommendations with fitness in male and female students are presented in Figure 1 and Figure 2, respectively. According to these analyses, active, public and private commuters, both men and women, who met MVPA recommendations reported higher general physical condition compared to active, public and private commuters who did not meet MVPA recommendations. In addition, male active and public commuters who met MVPA recommendations reported higher cardio-respiratory fitness, muscular strength, speed and agility and general health compared to students who use active and public commuting and did not meet MVPA recommendations. Finally, public commuters who met MVPA recommendations reported higher flexibility compared to those that use a public mode of commuting and did not meet MVPA recommendations. Additionally, women who use public and private modes of commuting and met MVPA recommendations reported higher cardio-respiratory fitness compared to those who use public and private modes of commuting and did not meet MVPA recommendations. Finally, public commuters who met MVPA recommendations reported higher muscular strength, speed and agility, flexibility and general health compared to those that use public commuting and did not meet MVPA recommendations.

## 4. Discussion

The main findings in the present study were that (a) the public mode of commuting was the most used and that men reported higher PA and fitness than women, (b) active and public commuters were more likely to meet the MVPA recommendations and reported higher muscular strength than those using private commuting and (c) active and public commuters who met MVPA recommendations present the highest fitness in most of its components.

Public commuting was the main mode of commuting to and from university for men and women. This mode of commuting may be beneficial for university students, as several studies have found. One study investigated an adult working population (over 18 years old) for 10 years and indicated that using public transportation services generally involves some walking (e.g., to bus stops or train stations) and could be an important contribution to energy expenditure [27]. In fact, a study in an English adult population (aged < 35 years old) showed that the public mode of commuting to and from work has an important contribution to PA levels, with 25.7 ± 14 min more per week of MVPA than private commuters [30]. Consequently, a key contribution of this research is the identification of differences in health and social implications between public and private modes of commuting. However, most of the studies to date identify both modes in the same category as passive modes [31]. In the present study, women had a lighter PA level, while men had a more vigorous PA level. In fact, a lower percentage of women reported that they were meeting MVPA recommendations compared with men. This finding is in agreement with studies in Spain and Colombia reporting that university women showed lower levels of PA compared with male students [32,33]. Similar results were already observed in Chile in 2013 among university students [34]. Therefore, it is necessary to implement programs promoting PA targeting specifically university women. Furthermore, in the current study, men showed higher mean scores in all perceived fitness components, except for flexibility, compared with female students. In line with our findings, similar results were obtained in a Slovenian study, with objective measures, where female university students presented better flexibility than men [17]. In this sense, a possible explanation for this might be that there are different stereotypes preconceived by society, such as that coordination in dance and/or higher flexibility has been associated mostly with women. Therefore, independent of the evaluation by a self-reported questionnaire, female participants may consider their flexibility as a high point of their fitness. Nevertheless, the rate of decline in flexibility with age was found to vary depending on the PA levels [35]. Therefore, if university women maintain their low PA levels throughout their adulthood, it is anticipated that their good flexibility scores could potentially decrease rapidly, and such stereotypes should come to an end.

Active commuters were more likely to reach higher moderate and vigorous PA levels (men and women, respectively) and meet the MVPA recommendations (both men and women) compared with those who used private commuting modes. These results are consistent with data obtained in adults, with positive associations between AC to work and PA [36,37], and a young population, where children and adolescents who use AC to school were significantly more physically active and were more likely to meet the MVPA recommendations than those who use private commuting [38], which reinforces the idea that AC may be a possible way to promote PA in the university population, as in any other population. Concerning fitness, a systematic review indicated that in the adult population, AC presents an important and positive association with fitness (cardio-respiratory fitness and muscular strength) [31]. In the present study, male active and public commuters showed higher muscular strength compared to those who used private modes of commuting. Similar results were reported in Norwegian children and adolescents (9 to 15 years) [39] and Finnish male young adults (25.5 ± 5.0 years) [40], where AC was positively associated with muscular strength. However, studies in adolescents and adults indicated that participants who use the cycling commuting present higher muscular strength, a benefit that appears to be greater than that achieved by walking commuters, because it will depend on the intensity [41,42,43]. In addition, female and male public commuters reported higher flexibility and general health levels, respectively, compared to students to those who used private modes of commuting. The lack of studies related to this topic in university students makes a comparison with the results of the present study difficult. Further research concerning the modes of commuting to university and fitness is necessary.

Finally, we analyzed the full relationship between PA behaviors (e.g., mode of commuting and MVPA recommendations) and fitness. Overall, low perceived fitness is reported among university students who do not meet MVPA recommendations, regardless of the mode of commuting. In fact, male and female public commuters who met MVPA recommendations present higher fitness in all cases (e.g., general physical condition, cardio-respiratory fitness, muscular strength, speed and agility, flexibility and general health); likewise, male active commuters who met MVPA recommendations present higher fitness (except in flexibility) compared to those not meeting the MVPA recommendations. Although the literature indicates that AC has the potential to generate the recommended weekly volume of PA [44], and PA has been associated with fitness in university students [17], according to our results, meeting MVPA recommendations was more associated with higher perceived fitness than the mode of commuting chosen among university students. Consequently, achieving the PA recommendations is more relevant than adopting an active mode of commuting in order to have better fitness, and it seems that students do not consider that AC could be a way to come closer to complying with these recommendations, as they see it as lighter behavior. In fact, despite the information available in the literature, AC could be undervalued and considered as light PA, but it should be considered as a potential behavior with relevant contributions to the physical health status and other associated benefits.

These findings provide evidence that a more physically active profile and a greater self-perception of fitness could be observed among more active commuters among university students. This indicated that new policies and public health actions could be developed to strategically target the increase in PA levels through the promotion of AC to and from university. However, further studies are required for a deeper examination of the relationships between PA levels, fitness and AC in university students, particularly studies using device-based measurement of PA and examining the contribution of AC modes to the PA recommendation. These findings could be applied in future programs to promote PA in the Chilean university population by encouraging the use of active commuting and assessing fitness and its implication on health.

### Limitations and Strengths

This study presented several limitations. The number of participants included was not representative of the entire Chilean university student population. In addition, the use of questionnaires to assess the mode of commuting, PA levels and fitness could be associated with recall bias and social desirability, and the use of device-measured physical activity (such as accelerometers) could have clarified the results of this study. On the other hand, a main strength of this study is the differentiation between public and private modes of commuting in our analysis, considering that most of the studies combine both modes as a unique mode called “passive”. In addition, it is important to highlight that to the best of our knowledge, this is the first study analyzing the relationship between the mode of commuting, the PA levels and the perceived fitness in Chilean university students, which represent an understudied population in the field of PA and health.

## 5. Conclusions

This study presents needed new evidence for the university student population regarding mode of commuting, PA and fitness. Public commuting was the main mode of commuting to and from university. Furthermore, active commuters were more likely to meet the MVPA recommendations and reported higher muscular fitness. In addition, low perceived fitness was reported among the university students who did not meet MVPA recommendations, regardless of the mode of commuting. Overall, the results highlighted the need for the development of actions or policies promoting PA specifically targeting university students. Further research targeting a broader understanding of the mode of commuting, PA levels and fitness in university students is needed.

## Figures and Tables

**Figure 1 healthcare-10-00990-f001:**
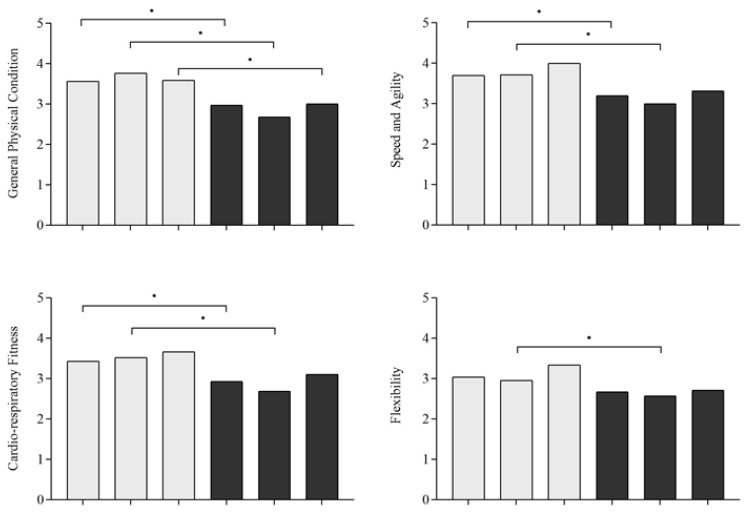
Associations of mode of commuting and meeting MVPA recommendations with fitness in men. Notes: active, public, private = mode of commuting; MVPA = Moderate to Vigorous Physical Activity; Meeting or Not meeting MVPA = MVPA recommendations; * = significant association with *p* < 0.05 between categorical groups.

**Figure 2 healthcare-10-00990-f002:**
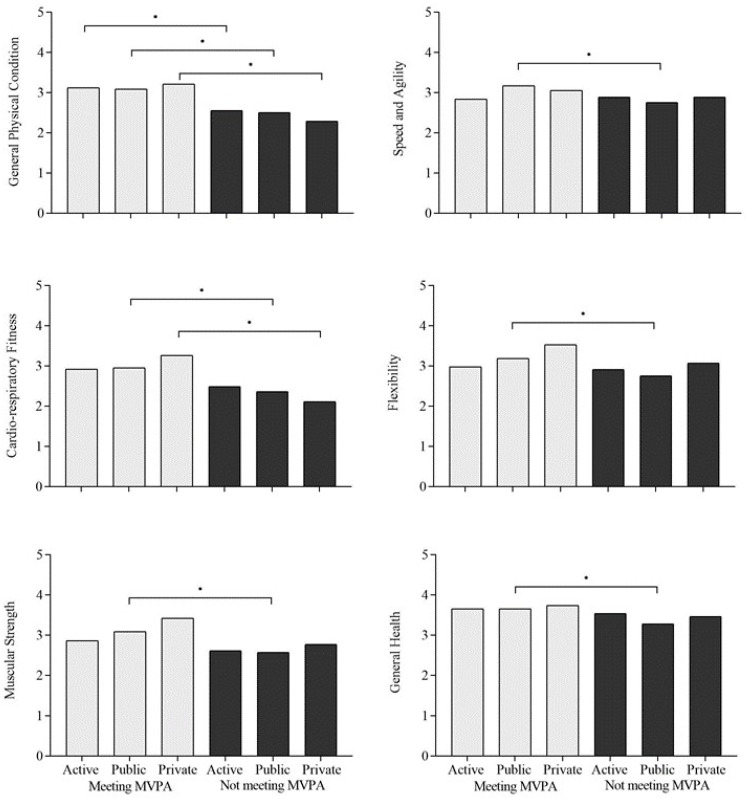
Associations of mode of commuting and meeting MVPA recommendations with fitness in women. Notes: active, public, private = mode of commuting; MVPA = Moderate to Vigorous Physical Activity; Meeting or Not meeting MVPA = MVPA recommendations; * = significant association with *p* < 0.05 between categorical groups.

**Table 1 healthcare-10-00990-t001:** Descriptive data of the mode of commuting, PA and sedentary behavior and fitness of the participants, and significant differences in the gender.

	All (*n* = 1257)	Men (*n* = 598)	Women (*n* = 659)	*p*-Value
**Mode of Commuting ***				
Active	331 (26.3)	206 (34.4)	125 (19.0)	<0.001
Public	791 (62.9)	332 (55.5)	459 (69.7)	
Private	135 (10.7)	60 (10.0)	75 (11.4)	
**PA and Sedentary Behavior**				
PA Levels *				
Light	482 (38.3)	155 (25.9)	327 (49.6)	<0.001
Moderate	365 (29.0)	184 (30.8)	181 (27.5)	
Vigorous	410 (32.6)	259 (43.3)	151 (22.9)	
MVPA recommendations *				
Not meeting	701 (55.8)	272 (45.5)	429 (61.2)	<0.001
Meeting	556 (44.2)	326 (54.5)	230 (41.4)	
Total PA (h/day) **	1.5 ± 1.7	1.9 ± 1.8	1.2 ± 1.4	<0.001
Sedentary behavior (h/day) **	6.5 ± 6.4	6.8 ± 6.9	6.2 ± 5.9	<0.001
**Fitness ******				
General physical condition	2.9 ± 1.0	3.2 ± 0.9	2.7 ± 0.9	<0.001
Cardio-respiratory fitness	2.8 ± 1.0	3.1 ± 1.0	2.5 ± 1.0	<0.001
Muscular strength	2.9 ± 0.9	3.2 ± 0.9	2.7 ± 0.9	<0.001
Speed and agility	3.1 ± 0.9	3.4 ± 0.9	2.7 ± 0.9	<0.001
Flexibility	2.8 ± 1.0	2.8 ± 1.0	2.9 ± 1.0	<0.001
General health	3.6 ± 0.9	3.8 ± 0.9	3.4 ± 0.8	<0.001

Notes: * = *n* (%); ** = X ± SD; X ± SD = mean ± standard deviation; PA = physical activity.

**Table 2 healthcare-10-00990-t002:** Associations of mode of commuting with PA, sedentary behavior and fitness of the participants by gender.

	Mode of Commuting to University *			
	Men		Women	
	Active	Public	Active	Public
	OR (95% CI)	OR (95% CI)	OR (95% CI)	OR (95% CI)
**PA and Sedentary Behavior**				
PA Levels				
Light	Ref.	Ref.	Ref.	Ref.
Moderate	**2.509 (1.14, 5.14)**	1.820 (0.86, 3.85)	1.315 (0.67, 2.54)	1.041 (0.59, 1.81)
Vigorous	1.478 (0.75, 2.90)	1.365 (0.72, 2.57)	**2.580 (1.14, 5.81)**	2.028 (0.98, 4.18)
MVPA recommendations				
Not meeting	Ref.	Ref.	Ref.	Ref.
Meeting	**1.351 (1.16, 2.04)**	1.229 (0.74, 2,24)	**2.031 (1.08, 3.81)**	1.577 (0.90, 2.74)
Total PA	0.898 (0.77, 1.04)	0.956 (0.83, 1.09)	**1.297 (1.01, 1.65)**	1.218 (0.97, 1.52)
Sedentary behavior	0.856 (0.32, 1.07)	0.703 (0.39, 1.24)	**0.301 (0.16, 0.54)**	**0.421 (0.25, 0.69)**
**Fitness**				
General physical condition	1.077 (0.80, 1.44)	1.217 (0.91, 1.61)	0.749 (0.55, 1.01)	0.813 (0.62, 1.05)
Cardio-respiratory fitness	1.154 (0.87, 1.52)	1.229 (0.94, 1.60)	0.784 (0.59, 1.03)	0.858 (0.67, 1.08)
Muscular strength	**1.691 (1.22, 2.33)**	**1.597 (1.17, 2.17)**	1.267 (0.94, 1.70)	1.214 (0.94, 1.56)
Speed and agility	1.205 (0.89, 1.62)	1.309 (0.98, 1.74)	1.074 (0.78, 1.46)	1.045 (0.80, 1.35)
Flexibility	1.134 (0.86, 1.49)	1.249 (0.96, 1.62)	1.256 (0.95, 1.65)	**1.295 (1.02, 1.63)**
General health	1.219 (0.87, 1.70)	**1.399 (1.01, 1.09)**	0.936 (0.67, 1.29)	1.171 (0.88, 1.54)

Notes: OR = odds ratio; 95% CI = 95% confidence interval; Ref. = reference; PA = physical activity; MVPA = moderate to vigorous physical activity; bold = significant association with *p* < 0.05; * = private commuting was established as reference.

## Data Availability

All data are available on paper in the laboratory of the IRYS research group at the School of Physical Education in Viña del Mar, Valparaíso, Chile.

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
