# Peer review of "Active Commuting to University Is Positively Associated with Physical Activity and Perceived Fitness"

_healthcare, 2022, doi:10.3390/healthcare10060990_

Round 1
Reviewer 1 Report
The article was very interesting to read.
It would be advisable to put the limitations in a separate section.
Author Response
We would like to thank the Editorial Committee and the Reviewers for their thoughtful and constructive comments and feedback.
Please find attached the answers to your comments.
Thank you very much.
The authors.

Reviewer 2 Report
General comment: The present study presents an interesting theme, but some aspects need to be improved throughout the text. Comments with these objectives follow below.
Comment: The introduction needs to be more well-designed. For example, what novelties does this study present when compared to studies previously in the literature? Where will the advances be identified?
“However, the association between fitness and AC in the 61 university student population has not been investigated”
Comment: This justification should not be the main point of this study in the introduction.
Comment: In the methods section, authors must present the sample size calculation.
Comment: How did the recruitment of the sample happen? How were participants invited? This information should be clearer in the methods.
Comment: How did the recruitment of the sample happen? How were participants invited? This information should be clearer in the methods.
“The self-reported questionnaire used for this study was created at the School of Physical Education by researchers with expertise on the topic of AC, called “Questionnaire of mode of commuting and PA to the university”. It was created based a literature review and consultations with experts. This questionnaire includes questions about sociodemographic variables and commuting behaviours. Every question was adapted to university student´s context and was found reliable for Chilean university students”
Comment: Has this questionnaire been validated? Did the authors perform any type of reproducibility to test this instrument?
Comment: In the evaluation of commuting PA, were the time spent commuting and the intensity practiced?
Comment: In the discussion is missing some mechanisms, for example, see the excerpt below::
“Furthermore, in the current study, men showed higher means score in all perceived fitness components, except for flexibility, compared with women students. Similar results were obtained in a study in a Slovenian study, where women university students presented better flexibility than men [17]. However, the rate of decline in flexibility with age was found to vary depending on the PA levels [35].
What are the reasons for these findings? Why did this happen? This information should close the idea of the paragraph.
“The lack of studies on this topic in university students challenges the possibility to make international comparisons and a future research demanding is to analyse it using objective PA measures”
Comment: Once again, the lack of this type of study in university students should not be the main justification. What are the innovative aspects of the present study when compared to previous studies? This should be clear from the discussion.
“ Overall, low perceived fitness is reported among the university students who do not meet MVPA recommendations, regardless of the mode of commuting”
Comment: Weren't these results already expected? What are the possible reasons for these results? It is necessary to deepen the discussion.
Comment: In the limitations, also insert that the different intensities of physical aiity were obtained through self-report, instead of using accelerometry, for example. The use of subjective measurement can cause bias in this type of information.
Comment: What are the practical applications of this study? This information should be entered at the end of the discussion.
Author Response

(The authors gave the same response as above.)

Reviewer 3 Report
The title is clear. The topic is current and important. These data and similar studies can help solve one extremely important problem. Abstract The summary is also clear and precise. It has all the necessary elements. Introduction The practical benefit of the results of this study should be added in the Introduction. Method I don't see the Design of study written anywhere, also the city and the university where the data will be collected. I WOULD RECOMMEND IT TO BE ADDED. It is interesting to me that the data were collected in different ways. In my opinion, that is more quality because the authors show imagination and experience. Following the explanation, it is not difficult to repeat the study. It is also a higher quality. the results The results are beautifully presented. Figures and tables are clear and precise. Discussion The study is interesting for the reader because the quality is higher again. It is recommended that the paper be published with minimal correction. References I THINK SOME REFERENCES NEED TO BE ADAPTED TO THE STANDARD. INTERNET STATEMENTS IN THE FIRST PLACE.
Author Response

(The authors gave the same response as above.)
